# Study of the Lamellar and Micellar Phases of Pluronic F127: A Molecular Dynamics Approach

**Juan M. R. Albano** [1,2], **Damian Grillo** [1,2], **Julio C. Facelli** [3], **Marta B. Ferraro** [1,2] and **Mónica Pickholz** [1,2,*]

1   Facultad de Ciencias Exactas y Naturales, Departamento de Física, Universidad de Buenos Aires, Buenos Aires 1428, Argentina

2   Instituto de Física de Buenos Aires (IFIBA), CONICET—Universidad de Buenos Aires, Buenos Aires 1428, Argentina

3   Department of Biomedical Informatics, University of Utah, 421 Wakara Way, Suite 140, Salt Lake City, UT 84108, USA

*   Correspondence: monicapickholz@gmail.com

**Abstract:** In this work, we analyzed the behavior of Pluronic F127 through molecular dynamics simulations at the coarse-grain level, focusing on the micellar and lamellar phases. To this aim, two initial polymer conformations were considered, S-shape and U-shape, for both simulated phases. Through the simulations, we were able to examine the structural and mechanical properties that are difficult to access through experiments. Since no transition between S and U shapes was observed in our simulations, we inferred that all single co-polymers had memory of their initial configuration. Nevertheless, most copolymers had a more complex amorphous structure, where hydrophilic beads were part of the lamellar-like core. Finally, an overall comparison of the micellar a lamellar phases showed that the lamellar thickness was in the same order of magnitude as the micelle diameter (approx. 30 nm). Therefore, high micelle concentration could lead to lamellar formation. With this new information, we could understand lamellae as orderly packed micelles.

**Keywords:** Pluronic F127; poloxamer; molecular dynamics; lamellar; micellar

## 1. Introduction

Poloxamers—commercialized as Pluronics (PL)—are linear nonionic triblock co-polymers of polyethylene oxide (PEO) and poly-propylene oxide (PPO). In Figure 1, a schematic representation is presented, where n is the number of hydrophilic ethylene oxide (EO) units, and m is the number of hydrophobic propylene oxide (PO) units. The length of each polymer block can be tuned to modify the poloxamer physical and chemical properties [1,2]. Many poloxamers have been approved by the Food and Drug Administration [3–5] to be used in pharmaceutical products [6–9]. In particular, Pluronic F127 (PL F127) is widely used in many applications due to its low toxicity, high drug loading capabilities, and ability to gel in physiological conditions at relatively low concentrations [7,10–13].

The phase behavior of the different PLs has been widely studied [14]. In particular, for PL F127, changes in temperature and concentration lead to micellization and gelation due to the transition from the liquid to the soft solid phase. This transition could be affected by the presence of drugs [13]. Besides, PLs can form lyotropic liquid crystals, exhibiting lamellar, hexagonal, or cubic phases [15]. It is difficult to access information on the internal organization of these system. In particular, for PL F127, the lamellar phase is commonly represented in a bilayer conformation, in our knowledge, without any experimental conformation [16]. This issue gains importance when considering the functionalization

of liposomes with this polymer [16]. Therefore, understanding PL F127 organization at the molecular level can help in the selection of the co-polymer composition for a given application.

Molecular dynamics (MD) simulations are a powerful tool to obtain structural information at the molecular scale. Due to the large size of PL F127, even for relatively simple systems, it is difficult to perform atomic-scale simulations. To bridge this gap, coarse-grain (CG) models are useful to accurately simulate these types of system, solving the size and time scale limitations of atomistic simulations [17,18]. Using these models in previous works, we were able to study the self-assembly, stability, and drug-loading capabilities of F127 micelles and their interaction with lipid bilayers [19,20].

In this work, we studied the molecular, structural, and mechanical properties of F127 micellar and lamellar phases through extensive molecular dynamics simulations from different initial polymer configurations using a CG model previously reported in the literature [19–21] for PL F127.

## 2. Methodology

In this work, we performed MD simulations at the CG level for Pluronic F127 using the Martini force field (MFF). The Martini CG model allows a systematic representation of molecules in terms of few building blocks [22]. Broadly, within MFF, four heavy atoms are represented in one site or CG bead. For PL F127, the parameters were taken from previous works [19–21]. In particular, detailed information of the used parameters could be found in the supplementary material of reference [19]. In all cases, polarizable water was used, where four water molecules are represent into one bead [23,24]. The parameters used are available at the MFF website (http://cgmartini.nl/index.php/force-field-parameters).

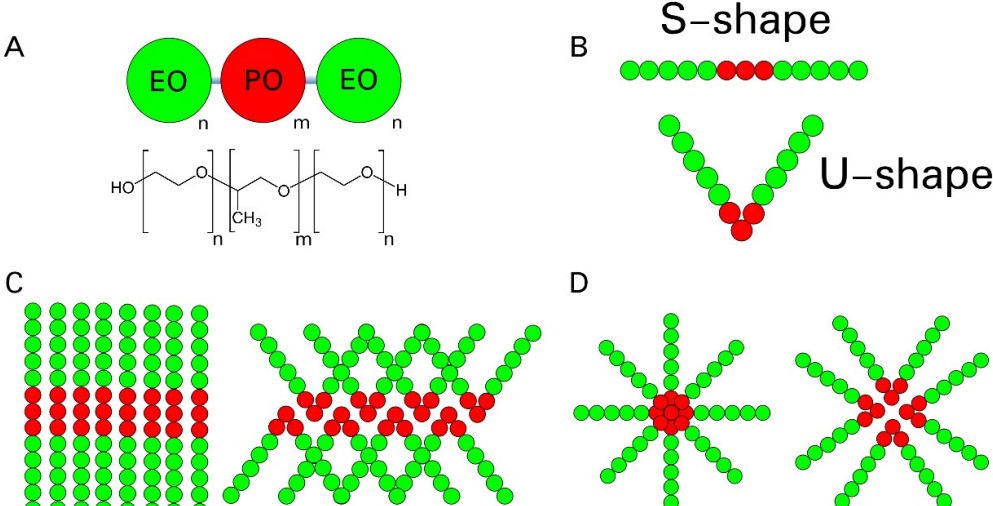

**Figure 1.** (**A**). Schematic representation of the coarse-grain (CG) model used in this work and chemical composition of Pluronics. (**B**). Representation of the F127 polymer in S-shape (top) and U-shape (bottom). Schematic representation of the (**C**) two lamellar (L) and (**D**) two micellar (M) cases explored in this work, composed of F127 polymers in S-shape (left, LS and MS) or U-shape (right, LU and MU) initial configuration. EO: ethylene oxide, PO: propylene oxide.

In Figure 1B, we present a schematic representation of the initial configuration of the simulated systems, S-shape and U-shape. Being PL F127 a linear triblock copolymer with two hydrophilic tails, the lamellar (L) phase could be formed by an extended monolayer (here called S conformation in the LS case) or a bilayer with the copolymer molecules directing their tails to the same water phase (U conformation in the LU case), as shown in Figure 1C. In Figure 1D, and a schematic representation of the micellar (M) system in both initial S (MS) and U (MU) polymer conformation is shown. All systems were pre-assembled into micellar o lamellar phases using the packmol software [25]. A summary of the four simulated systems discussed in this work is presented in Table 1. In this table, the term U

identifies the systems where the polymer started in a U-shape configuration (MU and LU), whereas S is used for the systems where the polymer started in a linear, straight, S-shape configuration (MS and LS).

**Table 1.** Summary of the four simulated systems considered in this study. In the second column (Initial), the initial configuration of the polymer at the beginning of the simulation is reported. F127 indicates the number or polymer used in the system, and PW the number of water beads considered.

| Structure | Initial | Case | F127 | PW | Total Beads | Run Time |
|-----------|---------|------|------|-----|-------------|----------|
| Micelle | U | MU | 100 | 591,204 | 617,704 | 3 μs |
|         | S | MS | 100 | 593,910 | 620,410 | 3 μs |
| Lamella | U | LU | 100 | 191,500 | 218,000 | 7 μs |
|         | S | LS | 100 | 191,500 | 218,000 | 7 μs |

It is important to remark that atomistic systems equivalent to the largest cases studied here (MU and MS) would consist of approx. 7,127,000 atoms, being most of them water molecules atoms (approx. 7,095,000). Even for simpler systems (LU and LS), all atom equivalents are challenging to simulate (approx. 2,300,000 atoms). As an example, recently, with the same high-performance computing resources, we carried out atomistic simulations of systems one order of magnitude smaller than the LU and LS systems [26]. Within this context, we were able to reach 0.5 μs of simulation run [27,28]. In the present study, the atomistic treatment for the kind of systems considered is hindered by their size, and a CG scale is a suitable alternative approach.

MD simulations were performed using the GROMACS 2018 package [29]. All simulations were carried out using the NPT ensemble (isotropic for micellar and semi-isotropic for lamellar systems), periodic boundary conditions (PBC) in all directions, shifted Lennard–Jones (LJ) potential (cutoff radius of 1.2 nm), shifted Columbic potential (cutoff radius of 1.2 nm). A global dielectric constant of $\varepsilon r = 2.5$ was set to ensure a realistic dielectric behavior of the hydrophobic regions using the polarizable water model [23]. The time step was of 20 fs, the temperature was equilibrated at 300 K using the Nose–Hoover thermostat [30] with a coupling constant of 6.0 ps, and the pressure was kept at 1 bar using the Parrinello–Rahman barostat [31] with a coupling constant of 6.0 ps and compressibility of $4.5 \times 10^{-5}$ bar$^{-1}$. The geometry of the water molecules was held fixed by means of the LINCS algorithm [32].

Density profiles are an appropriate measurement for the structural characterization of these systems [19,20,33–35]. For the micellar systems, the radial mass density profiles (rMDPs) were computed by dividing the systems into spherical shells along the radial direction centered at the micellar hydrophobic core and calculating the average density on the respective shells. For the lamellar systems, the mass density profiles (MDPs) were obtained by dividing the systems in thin slabs along the normal (z) direction and calculating the average density on the respective slabs. Pressure profiles are a measurement to get insights into the mechanical behavior of these types of system at the molecular level; different methods, depending on the system geometry, have been developed to compute them. For spherical systems, we used the method developed by Nakamura et al. [36] and for lamellas, we divided the systems in slabs along the z-direction (similarly to the MDPs) and computed tangential pressure (PT)(z) and normal pressure (PN)(z) for each slab. Details of the calculation method for PT(z) and PN(z) can be found elsewhere [37–39]. The calculation of the pressure profiles for both planar and spherical systems was implemented in GROMACS 2018 and its freely available at https://gitlab.com/damgrillo/gromacs-lpressure; it will be further discussed in a future publication.

## 3. Results and Discussions

### 3.1. Micellar Phase

In this section, we compare the structural and mechanical properties of two pre-assembled PL F127 micelles, MS and MU. The inner organization of the micelles can be accessed by the rMDPs.

In Figure 2A,B, we present the rMDP as a function of the radius for both micelle simulations. Each group is plotted separately, together with the whole system. Both micelles contain a core with no water access. From these figures, it is possible to see than inside the 6 nm sphere, both PPO and PEO are present with little differences between the MU and MS simulations. A PEO crown water interface and water phase is observed for both micelles at ~6 nm. The main difference between MU and MS simulations is that the interfacial PEO distribution exhibits a more pronounced peak in the S case. This feature leads to a more compact structure for the S case due to PEO accumulation.

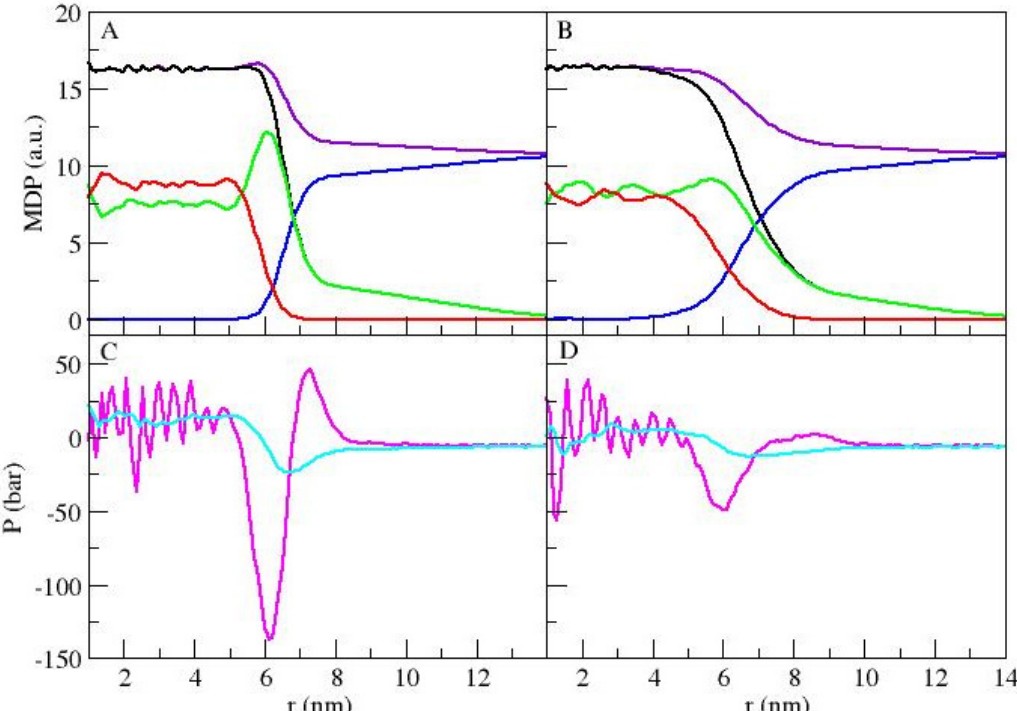

**Figure 2.** Mass density profiles (MDPs) and pressure profiles as functions of the radius for MS (**A**,**C**) and MU (**B**,**D**). For (**A**,**B**), total density (violet), total polymer (black), water (blue), poly-propylene oxide (PPO, red) and polyethylene oxide (PEO, green) are depicted. For (**C**,**D**), the normal (cyan) and tangential (magenta) pressure profiles are shown.

The normal and tangential pressure profiles for both systems are shown in Figure 2C,D. The systems show similar pressure profiles, but with different magnitudes. For the tangential pressure, it is possible to see large fluctuations for radius lengths less than 5 nm, followed by a minimum around 6 nm associated with the region where PPO density drops. The MS case minimum is about 2.5 times deeper than that of the MU case. The PEO water interface is characterized by a pressure peak, broader and shorter for the MU case. The maxima are located at ~7 nm and ~8 nm for the MS and MU cases, respectively, accounting for the extended micelles in the latest case. The normal pressure profiles also show similar patterns. For the MS system, we can see higher pressure amplitudes for both components, indicating higher accumulated stress at this interface. This can be related to the initial condition from which each polymer started. Being the S-shape polymer crossed over the center of the initial MS micelle, it is possible that these polymers keep memory of their initial condition. Self-assembled micelles, as obtained in experiments, are expected to have low quantities of S-shaped polymers due to the coil conformation of the copolymer in water [19]. However, the differences between S-shape and U-shape conformations could be important when controlling micelle preparation. Besides, this issue is more crucial for lamellar cases.

### 3.2. Lamellar Phase

In this section, we discuss the results of the lamellar systems, LS and LU. A comparison of the total energy and volume did not show any significant differences between them (as shown in Table 2). The total system area of the LS case was wider than that of the LU case (see Figure 3A and Table 2). This corresponds to the 40 nm reduction of the system box in z direction (Lz) of LS with respect to LU, (see Figure 3B and Table 2). The average values are also reported in Table 2. Considering that the volume of both cases was similar, a different overall organization of them was expected.

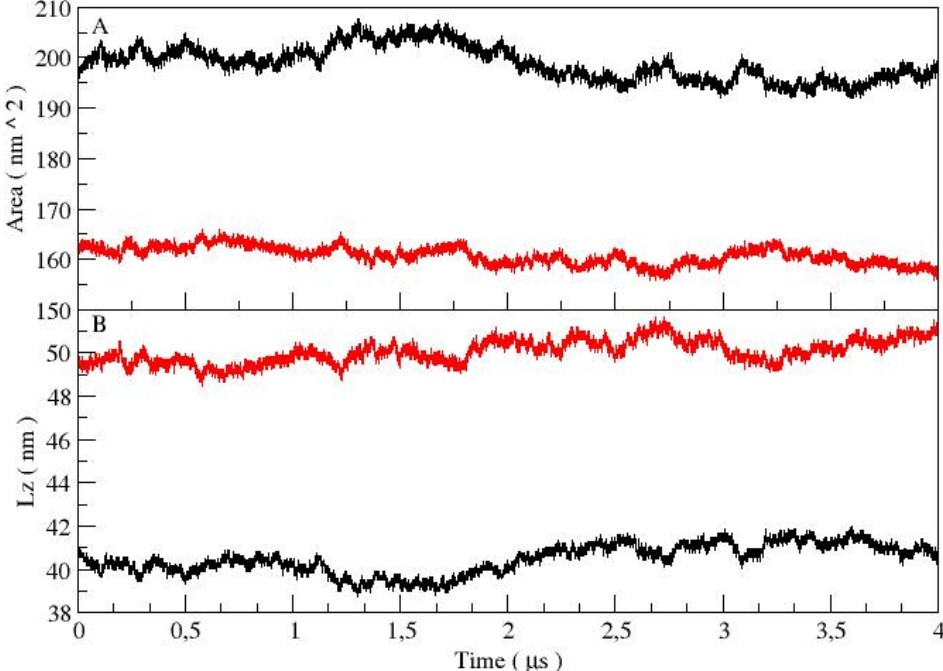

**Figure 3.** Total system area (**A**) and box in z direction (Lz) (**B**) are shown for both lamellar systems. In both figures, LS and LU are in black and red, respectively.

**Table 2.** Summary of calculated properties for both S and U lamellar systems.

| Property | LS | LU |
|---|---|---|
| Energy (kJ/mol) | −1,762,565 (3093) | −1,762,192 (3043) |
| Volume (nm$^3$) | 8045 (6) | 8041 (5) |
| Area (nm$^2$) | 198.8 (3.4) | 160.9 (1.9) |
| Lz (nm) | 40.5 (0.7) | 50.1 (0.5) |

The MDPs of total co-polymers and water are shown in Figure 4A for both cases. In this figure, three well-defined regions are evident: bulk water, polymer–water interface, and a core with no water. A comparison between the results of the LS and LU simulations showed some differences: the LS case presented a narrower thickness than the LU case, in agreement with the results of the Lz box size. The thickness difference between the two cases was approx. 2 nm.

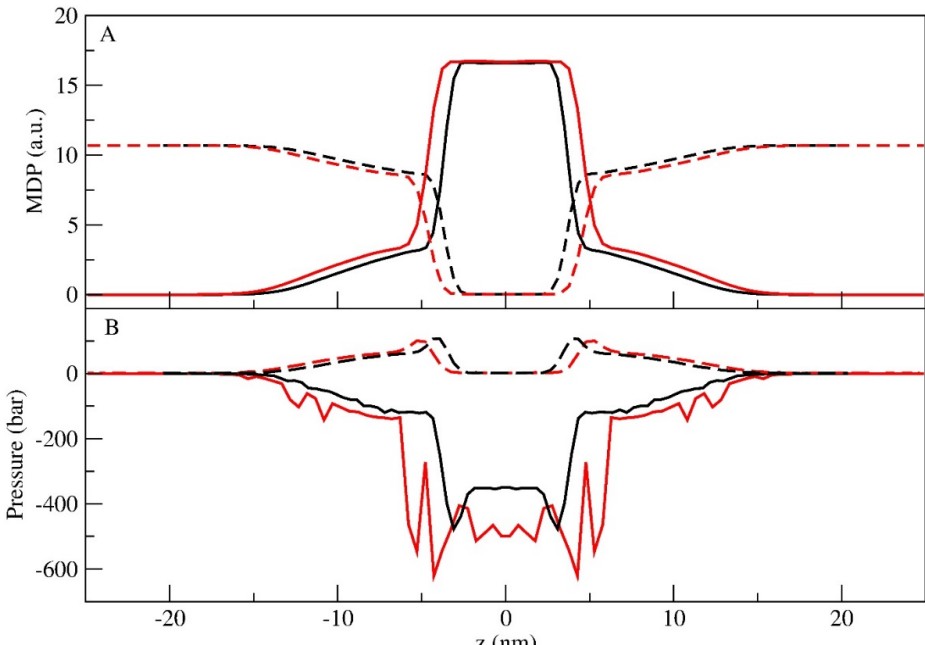

**Figure 4.** MDPs (**A**) and tangential pressure profiles along the z direction (**B**) for the total polymer (full line) and water (dash line). In both figures, the LS (black) and LU (red) simulations are presented.

Figure 4B shows the tangential pressure profiles of the co-polymers and water. The water tangential pressure profiles show similar patterns, and their differences could be correlated to the interface shifted distributions. On the other hand, the polymer tangential pressure profiles are more complex. For the LS simulation, as the system shifts to the center of the core, the pressure drops smoothly, presenting two distinctive minima at the polymer–water interface. However, for the LU case, the tangential pressure distribution at the polymer–water interface is more complex, showing large oscillation in this region.

In Figure 5A, we present the mass density profiles of PEO and PPO separately. Both density profiles are narrower for the LS case and wider for the LU case. Furthermore, it is possible to identify a differential distribution of PEO and PPO. While PPO mainly presents a homogenous central distribution, for PEO there are two distinctive peaks at the water–PEO interface. The PPO tangential pressure profiles for LU present two low-pressure minima relative to the PEO–PPO interface, while for the LS case, this is subtler as shown in Figure 5B. The most remarkable differences are present when observing the PEO pressure profiles of the two systems (Figure 5C). While, for LS–PEO, the tangential pressure progressively drops towards the center of the core with two distinct peaks at the PEO–water interface, for LU–PEO, the same high oscillations present for the total polymer can be observed. On the basis of these results, we can attribute a major role to the tangential pressure in the interactions between PEO and water.

The single polymer analysis was carried out through the combination of two tools: MDPs and trajectory analysis. First, we looked at the MDP profile of each polymer to capture the time average distribution. With this tool, it is possible to differentiate between the initial configurations. To define the S criteria, we considered that the MDP should have a density outside the range of −5 nm and 5 nm, corresponding to the lamella core. Besides, we followed the trajectory of three specific beads (the two terminal PEO beads and the central PPO bead) during the simulation time. For the LS case, the trajectories of the two PEO terminals should be outside the core in opposite sides. An example of a polymer that satisfies both conditions is shown in Figure 6A,B. For the LU single-polymer organization, we considered similar criteria for the core, considering that the end bead trajectories should be outside the core, on the same side. As an example, we show in Figure 6C,D the MDP and trajectories characteristics of a U shape co-polymer. The analysis performed for all co-polymers in the

systems showed that only a small percentage of the molecules remained in pure S or U conformation (20%). Also, we observed that all single co-polymer chains had a memory of their initial configuration, since no transition between S and U shapes was observed. Nevertheless, most copolymers had a more complex amorphous structure, where the PEO beads were part of the lamellar-like core.

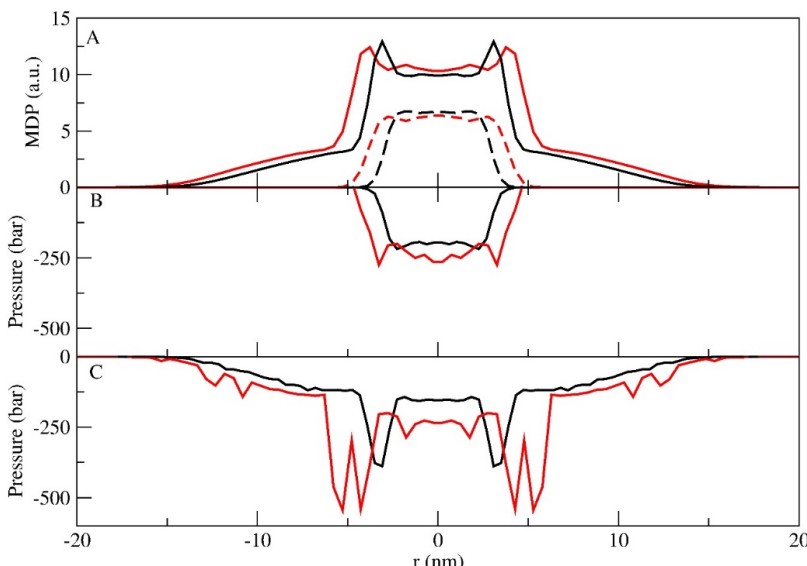

**Figure 5.** MDPs (**A**) for PEO (full line) and PPO (dash line). Tangential pressure profiles along the z direction for PPO (**B**) and PEO (**C**). In all figures, the LS and LU are shown in black and red, respectively.

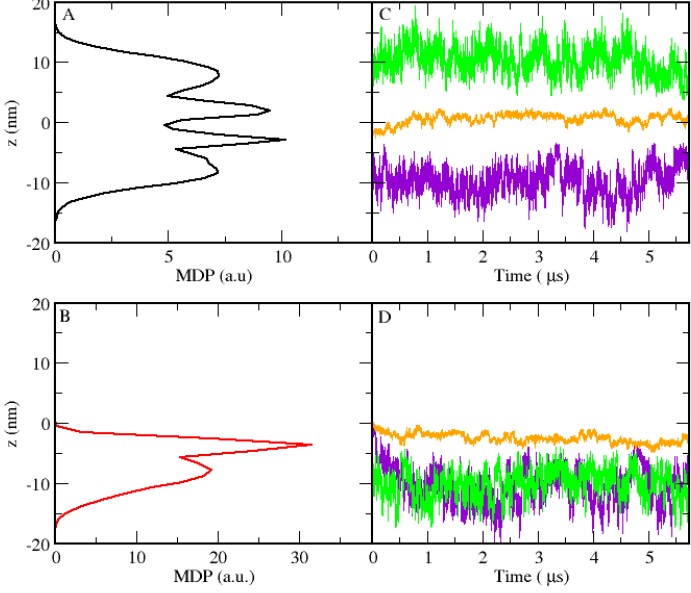

**Figure 6.** MDPs of a single polymer in the LS (**A**) and LU (**B**) cases, as functions of the z coordinate (rotated). Trajectories of the terminal PEO beads (green and violet) and the central PPO bead (orange) as functions of time for the same polymer of LS (**C**) and LU (**D**).

## 4. Conclusions

As a first step in this work, we compared the results of extensive MD simulations, at the CG level, of PL F127 micelles, starting from two different initial conditions (MS and MU). We noticed that the PPO and PEO distributions at the core of the micelles were similar. However, when looking at the interface, key differences arose. For the MS case, a distinctive drop of the pressure was observed at the PPO–PEO interface, followed by a positive pressure peak at the PEO–water interface. In the MU

case, this was subtler. This characteristic could influence drug partition in PL F127-based nanocarriers. Besides, this interface could also change with the pH, depending on a drug ionization state [13,37].

In the literature, it was reported that PL F127 could form lamellar phases [15], but information of the inner structural organization of lamellar phases is difficult to obtain using experimental techniques. Starting from two different initial conditions (LS and LU), we were able to find differences between them through the analysis of their structural and mechanical properties. We found that the studied lamellar properties were strongly influenced by the initial conformation. Under the described conditions, we did not observe transitions between U and S shapes in either direction during the simulation time. Nevertheless, different from what happens with lipid bilayers and other polymer-based systems [33,40], an amorphous core was present, where PPO and PEO beads coexisted. In Figure 7, representative snapshots of the two lamellar systems where the complex core structure can be observed are presented. Moreover, even if the polymers did not present the transition criteria, it is possible to notice that PEO was part of the lamellar core in both cases. Nevertheless, the size and flexibility of PL F127 prevented a whole PEO tail to cross through a very packed core (necessary for the U–S transition). U–S transitions are expected to be rare events, essentially driven by entropy. In the particular case of PL F127, specific water–polymer interactions (such as hydrogen bonds) could play an important role in these transitions. In coarse-grained simulations, these interactions affect free-energy barriers. In this way, the transition state energy evaluation could be challenging and not necessarily subject to cooperative events. Being out of the scope of this work, further exploration of this issue will be addressed in the future.

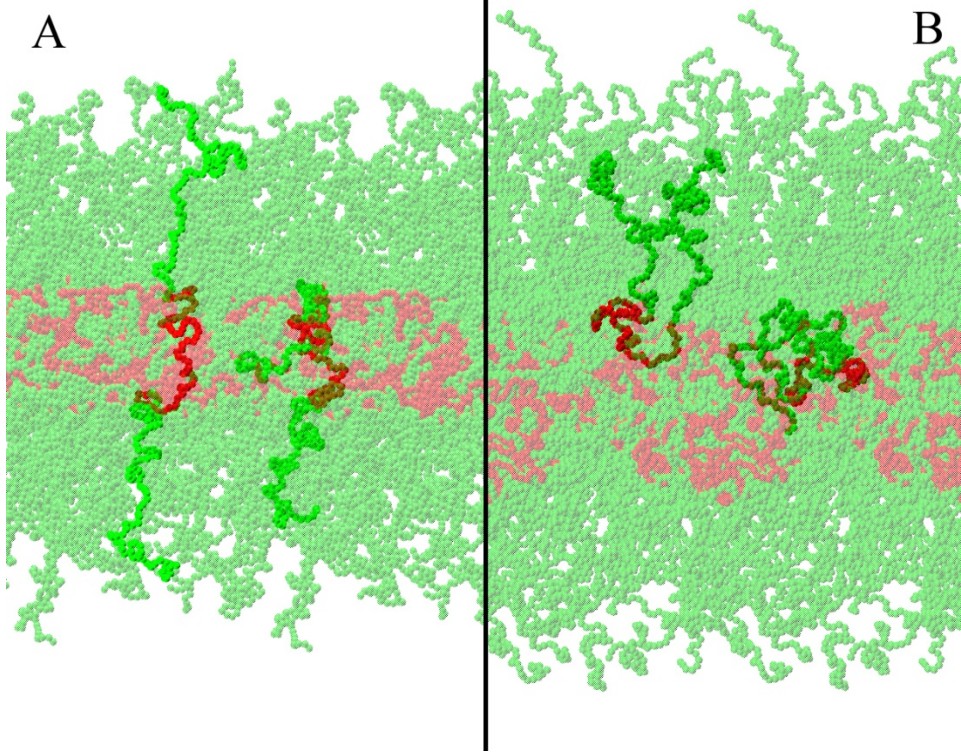

**Figure 7.** Snapshot of the LS (**A**) and LU (**B**) cases at the end of the simulation. For each snapshot, two individual polymers are highlighted, one representative of the S (**A**) or U (**B**) polymer configuration, and the other with a complex amorphous structure, where PEO beads are part of the lamellar-like core.

An overall comparison of micellar and lamellar phases discussed in this work showed that the lamellar thickness was in the same order of magnitude as the micelle diameter (approx. 30 nm). This is in good agreement with SAXS reported thickness [15]. In this way, a high micelle concentration could lead to lamellae formation. With this new information, we could understand lamellae as orderly packed micelles.

**Author Contributions:** Conceptualization, J.M.R.A. and M.P.; methodology, D.G., J.M.R.A., and M.P.; software, D.G.; validation, D.G., J.M.R.A., and M.P.; formal analysis, J.M.R.A. and M.P.; investigation, J.M.R.A. and M.P.; resources, J.C.F.; data curation, D.G., J.M.R.A., and M.P.; writing—original draft preparation, J.M.R.A., M.P.; writing—review and editing, D.G., M.B.F., and J.C.F.; visualization, D.G., J.M.R.A., and M.P.; supervision, M.P.; project administration, M.P.; funding acquisition, M.P., J.C.F., and M.B.F.

**Funding:** MF was funded by UBACYT 20020170100456BA, from Universidad de Buenos Aires; PIP 11220130100377, from CONICET. The Center for High-Performance Computing at The Utah University provided computer resources for High-Performance Computing, which was partially funded by the N.I.H. Shared Instrumentation Grant 1S10OD021644-01A1. J.C.F. was partially supported by the University of Utah Center for Clinical and Translational Science under NCATS Grant U01TR002538. M.P. was supported by CONICET No. 0131-2014 - 0131-2014 and ANPCyT No. PICT-2014-3653.

**Conflicts of Interest:** The authors declare no conflict of interest. The funders had no role in the design of the study; in the collection, analyses, or interpretation of data; in the writing of the manuscript, or in the decision to publish the results.

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
