# Peer review of "Study of the Lamellar and Micellar Phases of Pluronic F127: A Molecular Dynamics Approach"

_processes, doi:10.3390/pr7090606_

Round 1

Reviewer 1 Report

The authors present a study of the Lamellar and Micellar Phases of Pluronic F127.

They used the molecular dynamics methods to analyze this study. I think the presented analysis is interesting, but there are a number of issues with the paper in the present form.

Major points:

Which the coarse grain force filed is used in the M.S. ? Pls describe the parameters detailly. What’s the condition of pH in the simulations. The condition of pH can affect the formation of micelle, such as DOI: 10.1021/la304836e … Pls describe the setting of the simulation system detailly. How many water molecules …. Pls show the snapshots of lamella formation process.

Author Response

Comments and Suggestions for Authors

The authors present a study of the Lamellar and Micellar Phases of Pluronic F127.

They used the molecular dynamics methods to analyze this study. I think the presented analysis is interesting, but there are a number of issues with the paper in the present form.

Major points:

Which the coarse grain force filed is used in the M.S. ? Pls describe the parameters detailly.

The information about the used model and parameters was expanded (see lines 64-70)

 What’s the condition of pH in the simulations. The condition of pH can affect the formation of micelle, such as DOI: 10.1021/la304836e …

In Figure 1 we have incorporated the chemical structure of the poloxamer. The non-ionic nature of these polymers makes them independent of pH. However when they are used as drug delivery system the micellar structure could be affect by the protonation state of the drug, as the reviewer suggested. We have incorporated this in the manuscript (lines 42-43 and 230-231).

Pls describe the setting of the simulation system detailly. How many water molecules ….

This information was added (lines 69-70 and 92-94)

Pls show the snapshots of lamella formation process.

We started from two pre-assembled lamellas, as shown in Figure 1. We have included information about the building process in the manuscript (lines 82-83). Furthermore, in Figure 7 we have included representative snapshots of the simulations at their final state.

Reviewer 2 Report

Dear editor,

I carefully read the manuscript entitled "Study of The Lamellar and Micellar Phases of Pluronic F127: A Molecular Dynamics Approach". The topic of this study fits the scopes of 'Processes' journal and the overall results may interest people who are working in this field. However, I think several modifications should be made before publication. Here are my remarks:

Page 1, lines 30 to 38 (P1_L30-L38): For the sake of clarity, molecular schemes are required to illustrate the polyethylene oxide and poly-propylene oxide.

P2_L50: The authors state that molecular dynamics simulations are not possible at the atomic scale for their systems. It is hard to be convinced without the number of required particles for an atomic simulation... Nevertheless, such assertion must be more deeply argued. 

P3_L93: A reference is missing for this sentence.

P8_L215: The results and analyses made by the authors are well-mastered and the conclusions are meaningful. However, the observation that no pathway was observed between LS and LU lamellar structures was obviously expected. Indeed, with coarse grain simulations such events is hardly accessible since entropy is weakly mastered in CG dynamics. The authors should comment this strong limiation of their CG model. Moreover, enquiries along a collective variable which decipher the LS-LU transition should provide this pathway and evaluate the transition state energy. I know that these investigations are not easily accessible and not in the scope of this article, but I think the authors must, at least, comment this point.

Author Response

I carefully read the manuscript entitled "Study of The Lamellar and Micellar Phases of Pluronic F127: A Molecular Dynamics Approach". The topic of this study fits the scopes of 'Processes' journal and the overall results may interest people who are working in this field. However, I think several modifications should be made before publication. Here are my remarks:

Page 1, lines 30 to 38 (P1_L30-L38): For the sake of clarity, molecular schemes are required to illustrate the polyethylene oxide and poly-propylene oxide.

 Done

P2_L50: The authors state that molecular dynamics simulations are not possible at the atomic scale for their systems. It is hard to be convinced without the number of required particles for an atomic simulation... Nevertheless, such assertion must be more deeply argued.

 Now we have included this discussion (lines 92-99)

P3_L93: A reference is missing for this sentence.

Done

P8_L215: The results and analyses made by the authors are well-mastered and the conclusions are meaningful. However, the observation that no pathway was observed between LS and LU lamellar structures was obviously expected. Indeed, with coarse grain simulations such events is hardly accessible since entropy is weakly mastered in CG dynamics. The authors should comment this strong limiation of their CG model. Moreover, enquiries along a collective variable which decipher the LS-LU transition should provide this pathway and evaluate the transition state energy. I know that these investigations are not easily accessible and not in the scope of this article, but I think the authors must, at least, comment this point.

This discussion is revisited in the conclusion (lines 232-249) and Figure 7.

Round 2

Reviewer 1 Report

After carefully reading this M.S., the authors have addressed my comments.